# Detailed Studies on the Methoxylation and Subsequent Dealkylation of *N,N*-Diethylbenzenesulfonamide Using a Tailor-Made Electrosynthetic Reactor

**DOI:** 10.3390/molecules29235496

**Published:** 2024-11-21

**Authors:** Ernák F. Várda, Imre Gyűjtő, Ferenc Ender, Richárd Csekő, György T. Balogh, Balázs Volk

**Affiliations:** 1Department of Chemical and Environmental Process Engineering, Budapest University of Technology and Economics, Műegyetem rakpart 3, H-1111 Budapest, Hungary; varda.ernakferenc@edu.bme.hu (E.F.V.); gyujtoimre@gmail.com (I.G.); 2Egis Pharmaceuticals Plc., Directorate of Drug Substance Development, P.O. Box 100, H-1475 Budapest, Hungary; 3Department of Electron Devices, Budapest University of Technology and Economics, Műegyetem rakpart 3, H-1111 Budapest, Hungary; ender.ferenc@vik.bme.hu (F.E.); csekorichard@edu.bme.hu (R.C.); 4Spinsplit Research and Development Ltd., Szőlőskert u. 0182/135, H-2220 Vecsés, Hungary; 5Department of Pharmaceutical Chemistry, Semmelweis University, Hőgyes Endre út 9, H-1092 Budapest, Hungary; 6Center for Pharmacology and Drug Research & Development, Department of Pharmaceutical Chemistry, Semmelweis University, H-1085 Budapest, Hungary

**Keywords:** electrochemistry, Shono oxidation, anodic oxidation, benzenesulfonamides, methoxylation, dealkylation

## Abstract

Benzenesulfonamides are an outstandingly important family of compounds in organic and medicinal chemistry. Herein, we report detailed studies on the electrochemical mono- and dideethylation of model compound *N,N*-diethylbenzenesulfonamide. In this context, all parameters of the electrosynthesis were systematically investigated, with a special emphasis on solvent screening and the effect of water on the outcome of the reaction. Beside a commercially available electrochemical reactor, a custom-made device has also successfully been designed and used in these transformations. Optimization of the reaction led to a green, scaled-up synthesis of the dealkylated products. Our experiments also render the synthesis and potential in situ use of the corresponding *N*-methoxyalkyl intermediate, a precursor of the reactive and versatile *N*-sulfonyliminium cation, possible.

## 1. Introduction

The benzenesulfonamide unit (Figure 1) is a common motif in several classes of drugs [1]. More than 70 drugs containing this structural unit are currently on the market for the treatment of various diseases. Among these, those *N,N*-dialkyl (tertiary sulfonamide) and *N*-monoalkyl (secondary sulfonamide) derivatives in which the nitrogen atom is not part of a hetero ring are the most relevant examples for the present study (Figure 1) [2,3,4,5,6,7,8,9,10,11,12,13,14,15,16,17].

It is important to highlight that one of the most important metabolic transformations of drugs containing an *N*(sp^2^)-*C*(sp^2/3^) structural unit is *N*-dealkylation [18], which is typically catalyzed by the cytochrome P450 enzymes [19]. In the first step, the enzymatic biotransformation begins with the abstraction of the H atom from the *C*(sp^2/3^) atom and the incorporation of a hydroxyl group, followed by the oxidative dealkylation to form end-products of metabolism containing *N*(sp^2^)-H and *C*(sp)=O motifs [20].

However, one of the important driving forces of oxidative metabolism is the lipophilicity of drugs [21,22]. There are only a few examples in the literature for the *N*-dealkylation of relatively polar *N*-(di)alkylated sulfonamides as a major metabolic pathway. In this context, among the drugs shown in Figure 1, only the major metabolic pathways of probenecid [23] and udenafil [24] are associated with the *N*-dealkylation process.

Besides common protecting group strategies (e.g., acidic cleavage of substituted *N*- or *O*-benzyl groups, *N*- or *O*-debenzylation via hydrogenation), in certain cases, alkyl groups can also be applied as orthogonal protecting groups (e.g., acidic cleavage of *t*-BuO groups). Some chemical methods (see Appendix A) have already been described for the selective *N*-dealkylation of secondary or tertiary benzenesulfonamides by cleavage of the C–N bond, including the removal of a propargyl group under ruthenium catalysis [25], silver(I)-catalyzed *N*-deprenylation [26], oxidative *N*-debenzylation in the KBr/Oxone^®^ system [27], and *N*-dealkylation with periodic acid catalyzed by chromium(III) acetate hydroxide [28]. Dealkylation may not stop at the secondary sulfonamide stage in the case of starting materials of the *N,N*-dialkylsulfonamide type.

The electrochemical behavior of benzenesulfonamides has been widely investigated in the emerging field of wastewater technologies aiming at the elimination of antibiotics [29] and in the development of electroanalytical methods [30]. Under reductive electrochemical conditions, mild desulfonylation of amines can be carried out as a deprotecting strategy [31,32,33,34]. On the other hand, Shono oxidation is an anodic electrochemical oxidation in the course of which a new bond is formed in the α-position of an amino moiety between the carbon atom and the nucleophile applied in the reaction [35]. The reaction mechanism is illustrated in Figure 1 on the example of carboxamides: they can be oxidized to *N*-acyliminium ions, and then these can be captured by various nucleophiles (Figure 1).

Banks and Jones have conducted significant work to draw the attention of chemists to Shono oxidation [36] and reported on its utility in drug metabolism studies [37]. They thoroughly investigated the electrochemical behavior of *N,N*-diethylbenzamide [38]. The constant voltage electrochemical method with an undivided cell resulted in an α-methoxylated Shono product using tetrabutyl ammonium perchlorate (Bu_4_NClO_4_, TBAP) as the electrolyte in a CH_3_CN/MeOH solvent mixture (Figure 2A).

Here, MeOH acted as the nucleophile. Changing both the electrolyte and the solvent afforded enamide as the sole product [38]. They have recently published a constant-current procedure for the selective monodeethylation of the same substrate (Figure 2A) [39]. Shono oxidation was also studied on *N*-benzenesulfonyl- and *N*-tosylpiperidines and pyrrolidines (Figure 2B) [40,41,42,43]. Here, mono- and di-α-methoxylated and enamide-type products were only described; no dealkylated product was isolated.

In a recent paper, Wetzel and Jones published the first oxidative electrochemical cleavage of the *N*(sp^2^)–*C*(sp^2/3^) bond of benzenesulfonamide derivatives, i.e., the removal of *N*-aryl or *N*-alkyl groups [44]. This study, carried out under both batch and flow conditions, has a twofold relevance: First, it serves as a mild deprotection method, and second, this reaction mimics the cytochrome P450-catalyzed dealkylation of sulfonamide-type drug candidates. In the batch experiments, reticulated vitreous carbon (RVC) electrodes, LiClO_4_ electrolyte, and a CH_3_CN/MeOH (9:1) solvent mixture have been used at room temperature in an undivided cell setup (Figure 3). Variously substituted benzenesulfonamides were investigated. After a passage of 4.0 F/mol of charge, selective monodealkylation or monodearylation was observed in most of the cases, with the only exception being a cyclic derivative, *N*-tosylpyrrolidine, where the dehydrogenative coupling product (i.e., the corresponding α-methoxy derivative) was isolated in 55% yield. If the monoalkyl or monobenzyl compounds were further treated with an additional charge of 4.0 F/mol, or 8.0 F/mol was directly applied for the starting *N,N*-disubstituted congeners, the products unsubstituted at the sulfonamide moiety (i.e., the ArSO_2_NH_2_ products) were obtained (Figure 3). In the absence of MeOH, no appreciable dehydrogenative coupling or dealkylation reaction took place.

## 2. Results

Using only electrons as the oxidant, the electrochemical dealkylation published by Wetzel and Jones is an inherently green procedure. Although their study provided valuable insight into the electrochemical behavior of benzenesulfonamides, some aspects remained unclear. We inquired how changing the solvent or solvent mixture would influence the outcome of the reaction and whether the targeted synthesis of the synthetically useful α-methoxy compound, as a versatile intermediate for further transformations, is possible.

In our experiments, *N,N*-diethylbenzenesulfonamide (**1**) was used as the model compound under batchwise, controlled-current electrochemical conditions in an undivided cell at room temperature. The preliminary studies were carried out in a 10-mL vessel (containing 6 mL solvent) using an IKA ElectraSyn 2.0 Pro reactor in MeOH or MeOH-containing solvent mixtures. The supposed isolable products (monitored by LC-UV at 220 nm and by LC-MS) are shown in Figure 4. It was presumed that the oxidatively coupled product (**2a**) is primarily formed, analogously to the mechanism depicted in Figure 1. After hydrolysis of *N,O*-acetal **2a**, monoethyl derivative **3** is obtained, which is then further transformed into another possible methoxy intermediate (**4**). The formation of the doubly dealkylated sulfonamide **5** as the final product was also expected. It is noteworthy, however, that in our experiments carried out under a wide range of conditions, compound **4** was never identified by LC-MS in the crude reaction mixtures, most likely due to its instability.

In the first set of experiments, the following parameters were varied: electrical input (10–100 mA), quantity of starting material 1 (20–50 mg), type of electrodes (RVC or graphite—G), charge (2.3–11.25 F/mol), solvent (MeOH or CH_3_CN/MeOH 9:1), and electrolyte (0.500 M TBAP or Bu_4_NBF_4_). The full set of results is summarized in the Appendix A, while some characteristic ones are shown in Table 1. Under these conditions, methoxy compound **2a** was the main component in most of the cases. The optimal conditions proved to be 20 mA, 0.500 M TBAP, graphite electrodes, and 35.2 mg (27.5 mM) starting material. It is to be noted that the relative product amounts of starting material **1**, those of identified products **2a**, **3**, **5**, and, in some experiments, those of further derivatives of unknown structure (reported in Table 1 and in all other tables) were determined by HPLC/UV. If the total amount of unidentified compounds was negligible (below 1%), these were disregarded. Above this limit, the HPLC peaks of by-products were also integrated. In order to reduce the amount of the electrolyte (TBAP) applied, its concentration was decreased to 0.167 M in the following experiments. Furthermore, a tailor-made electrosynthetic reactor was constructed and tested in comparison to the commercially available IKA reactor. The original IKA reactor vial, electrodes, and cell geometry remained the same, but a more straightforward multi-channel DC laboratory power supply was used, allowing for parallel testing under constant current or voltage conditions. The cap was modified to allow sampling with a standard 0–200 μL automatic pipette, which is simpler and more accurate than using a syringe and needle.

So that the reproducibility of the reactions can be assessed, the reactions were conducted twice using our own reactor and once using the IKA reactor for each parameter set. The results (see Appendix A) clearly indicate that the two reactors are interchangeable. The new equipment turned out to be cheaper, simpler, more efficient, and easier to supplement with further units (e.g., sampler). Further studies were therefore performed in this in-house-designed reactor. The detailed technical description of the reactor can be found in the Appendix A.

In the next set of experiments (for the full list, see Appendix A, selected reactions are demonstrated in Table 2), the effect of 1% water was tested on the outcome of the reaction. It was concluded that, using the optimal electrical input (20 mA) and charge (4.5 F/mol) values, the formation of methoxy intermediate **2a** was preferred in pure MeOH, while the formation of deethylated product **3** became slightly more significant in the MeOH/H_2_O 99:1 mixture. Formation of benzenesulfonamide **5** was not observed.

A detailed solvent screening was also carried out in various mixtures containing an alcoholic solvent [MeOH, iPrOH (IPA), or *t*-BuOH] mixed with H_2_O and/or CH_3_CN. When IPA or *t*-BuOH were used instead of MeOH, the main products generated during oxidative coupling were **2b** or **2c** (as shown in Figure 2), respectively, instead of **2a**.

The comprehensive solvent screening results can be found in Appendix A of the Supporting Information, and characteristic reactions are demonstrated in Table 3. The formation of methoxy intermediate **2a** was found to be preferred in MeOH/H_2_O 95:5 (4.5 F/mol) or CH_3_CN/MeOH/H_2_O 85:10:5 (4.5 F/mol) mixtures, while that of deethylated product **3** was predominant in CH_3_CN/IPA/H_2_O 85:10:5 (4.5 F/mol) or CH_3_CN/*t*-BuOH/H_2_O 85:10:5 (9.0 F/mol). In some reactions in these series, a significant amount of the doubly deethylated derivative **5** was also formed. The optimal solvent mixture for the complete removal of both ethyl groups was CH_3_CN/IPA/H_2_O 89:10:1 (9.0 F/mol). It is noteworthy that **2b** and **2c** were only detected in some reactions and in significantly lower quantities than methoxy congener **2a** (Appendix A).

We aimed to elaborate targeted scaled-up methods for the synthesis of compounds **2a**, **3,** and **5**, respectively, in a more concentrated solution, i.e., starting from 105.6 mg (82.5 mM) of starting material **1** in the same 10-mL vessels. The reactions were carried out in a MeOH/H_2_O 99:1 mixture, which proved to be more optimal in the scaled-up experiment, likely due to the increased concentration, than the 95:5 ratio determined above. As demonstrated by the experiments summarized in Table 4, using a charge of 3.0 F/mol, the conversion was 94%, and the relative quantity of **2a** reached a maximum of 92%. When the reaction was continued, a full conversion of starting material **1** was observed; however, a significant decomposition (resulting in the formation of UV-undetectable components) occurred, demonstrated by the reduction of the peak areas in the UV (λ = 220 nm) chromatogram. This explains the seemingly increasing quantity of starting material **1** after more than 2 h of reaction time.

Our efforts toward the targeted scaled-up synthesis of monoethyl derivative **3** are shown in Table 5. In the CH_3_CN/*t*-BuOH/H_2_O 85:10:5 solvent mixture applied, the conversion of **1** was practically complete after 4 h (6.0 F/mol), reaching a maximum quantity of 3 near 80%. Continuation of the electrosynthesis led to decreasing amounts of **3** and the formation of doubly deethylated product **5**. It is noteworthy that, contrary to reactions carried out in solvent mixtures containing MeOH (e.g., Table 4), in these cases, the potential *tert*-butoxy intermediate **2c** was not detected at all.

The scaled-up synthesis of deethylated product **5,** starting from 105.6 mg of **1,** was attempted in the previously (Table 3) used CH_3_CN/IPA/H_2_O 89:10:1 ternary solvent mixture. As demonstrated in Table 6, the amount of starting material **1** and isopropoxy intermediate **2b**, which was detected in all cases, became insignificant after ca. 4 h reaction time, but the quantity of intermediate **3** could not be suppressed as efficiently as in the reaction conducted with a lower concentration of 1 (see Table 3 and Appendix A).

Finally, the three products (**2a**, **3 [45,46]**, **5 [47,48]**) were isolated from scaled-up electrosyntheses (starting from 0.500 mmol of **1**, Table 7) by preparative thin-layer chromatography (TLC) or flash chromatography. HPLC monitoring of the conversion showed that an increased charge transfer was beneficial in the case of methoxy derivative **2a**: 10.0 F/mol instead of 3.0 F/mol (indicated in Table 4), although such a high charge transfer led to decomposition at lower starting material concentration (Table 4). Another deviation from the optimized conditions (Table 4) was necessary: Although the higher electrolyte concentration and the use of tetrabutylammonium tetrafluoroborate instead of TBAP, as well as the absence of water (i.e., use of pure MeOH) reduced the yield, it made the purification process easier, as fewer by-products were formed. In conclusion, while the preparation of **2a** was not described in the paper of Wetzel and Jones [44], we have successfully synthesized it with 55% yield and characterized it as a new compound, despite literature findings indicating that structurally similar benzenesulfonamide derivatives containing a methoxymethyl group (MOM) on the sulfonamide nitrogen atom are prone to hydrolysis at room temperature [49,50]. In the presence of Lewis acids, analogous MOM-protected benzenesulfonamides are described as precursors of *N*-sulfonyliminium cations, which, due to their highly electrophilic character, are useful building blocks in the synthesis of nitrogen-containing heterocycles [51,52]. This suggests the possible in situ use of *N,O*-acetal **2a**, formed in our reaction, for similar synthetic purposes.

Using 4.5 F/mol charge transfer rendered the simultaneous isolation of the two dealkylated products (**3**: 62%, **5**: 25%) possible from the same scaled-up reaction (Table 7). In the targeted synthesis of doubly dealkylated product **5**, an increased charge transfer was again the key factor: using 13.5 F/mol led to an isolated yield of 41% (Table 7).

## 3. Materials and Methods

In all experiments, 10-mL IKA vials (IKA Werke, Staufen im Breisgau Germany) and IKA graphite SK-50 or IKA RVC foam electrodes (IKA Werke, Staufen im Breisgau Germany) were used. The reactions were carried out in an IKA ElectraSyn 2.0 Pro (IKA Werke, Staufen im Breisgau Germany) or in a tailor-made reactor. The latter was built from the following units: Hameg Triple Power Supply, IKA RH basic 2 magnetic stirrer, and National Instruments myDAQ data collector (National Instruments, Austin, TX, USA). HPLC separations were carried out on a Kinetex XB C18 column (150 × 4.6 mm, 2.6 µm, Phenomenex, Torrance, CA, USA). The pump flow rate was set to 1.0 mL/min, and the column temperature was 40 °C. The composition of mobile phase A was 0.1% (*v*/*v*) formic acid in H_2_O, and mobile phase B was the mixture of CH_3_CN and H_2_O 80/20 (*v*/*v*) with 0.1% (*v*/*v*) formic acid. The applied gradient program was the following: between 0–11 min, a linear gradient of 5–100% B was applied, and then B was kept at 100% between 11–13 min. After 13 min, B dropped to 5%, and 2 min of equilibration period was applied prior to the next injection. The injection volume was set at 10 µL, and the chromatographic profile was registered at 220 ± 4 nm. HPLC-MS analysis was carried out on a Waters 2690 HPLC (Waters Corp., Milford, MA, USA) system equipped with a Waters 2487 dual λ absorbance detector coupled with a Waters Micromass Quattro Ultima Pt tandem quadrupole mass spectrometer equipped with an ESI source. The mass spectrometer detector (MSD) operating parameters were as follows: electrospray ionization (ESI) positive ionization, scan ion mode (*m*/*z* 150–500), drying gas temperature 350 °C, nitrogen flow rate 5.5 L/min, nebulizer pressure 6 bar, quadrupole temperature 120 °C, capillary voltage 2500 V, and fragmentor voltage 60 V. MassLynx 4.0 software was used for data acquisition and analysis.

Scaled-up synthesis of **2a**. Compound **1** (105.6 mg, 0.500 mmol) and tetrabutylammonium tetrafluoroborate (1195 mg, 1.21 mmol) were measured into a 10-mL vial, then MeOH (6 mL) was added. The reaction mixture was stirred with a magnetic stirrer until a clear solution was obtained. Then the lid was placed onto the vial with two SK-50 graphite electrodes, and the electrolysis was started. During the electrolysis, a 20 mA DC current was set, and the stirring rate was 400 min^−1^. After a 10 F/mol charge, the reaction was stopped, the clear solution was transferred into a 50-mL round-bottom flask, and the solvent was evaporated onto silica gel (2.50 g) under reduced pressure. Flash chromatographic purification of the pale yellow oily residue with hexane–EtOAc eluent gave product **2a** (67 mg, 55%) as a colorless liquid. IR (film, cm^−1^): 2990, 2940, 1447, 1336, 1181, 1152, 1055, 1017, 950, 735, 692, 595, 573. ^1^H NMR (600 MHz, DMSO-*d*_6_): 7.82 (m, 2H, HAr), 7.67 (m, 1H, HAr), 7.61 (m, 2H, HAr), 5.09 (q, J = 6.0 Hz, 1H, CH), 3.18 (m, 1H, CH_2_), 3.11 (m, 4H, OCH_3_ + CH_2_), 1.11 (m, 6H, CH_3_).^13^C NMR (150 MHz, DMSO-*d*_6_): 140.69, 133.00, 129.58, 126.80, 85.76, 54.87, 36.61, 19.58, 16.49. MS *m*/*z* = 266.0818 ([M + Na]+, 20%); 212.0740 ([M-CH_3_OH]+, 100%).

Scaled-up synthesis of **3**. Compound **1** (105.6 mg, 0.500 mmol) and TBAP (414 mg, 1.21 mmol) were measured into a 10-mL vial, then CH_3_CN (5.1 mL), *t*-BuOH (0.60 mL) and H_2_O (0.25 mL) were added and mixed with a magnetic stirrer until a clear solution was obtained. Then the lid was placed onto the vial with two SK-50 graphite electrodes, and the electrolysis was started. During the electrolysis, a 20 mA DC current was set, and the stirring rate was 400 min^−1^. After a 4.5 F/mol charge, the reaction was stopped, and the reaction mixture was evaporated to half its volume under reduced pressure. Then it was moved into a 100-mL beaker, TBAP was precipitated upon the addition of diethyl ether (50 mL) and filtered through a cotton pad. The clear pale-yellow solution was transferred into a 50-mL round-bottom flask in portions and the solvent was evaporated under reduced pressure. Preparative TLC (hexane–EtOAc 1:1) purification of the brown semi-solid residue gave product **3** (57.4 mg, 62%) as a colorless liquid and **5** (19.7 mg, 25%) as a white solid. The ^1^H NMR spectra of compounds **3** [45,46] and **5** [47,48] were in accord with data from the literature.

Scaled-up synthesis of **5**. Compound **1** (105.6 mg, 0.500 mmol) and TBAP (414 mg, 1.21 mmol) were measured into a 10-mL vial, then CH_3_CN (5.35 mL), IPA (0.60 mL) and H_2_O (50 μL) were added and mixed with a magnetic stirrer until a clear solution was obtained. Then the lid was placed onto the vial with two SK-50 graphite electrodes, and the electrolysis was started. During the electrolysis, a 20 mA DC current was set, and the stirring rate was 400 min^−1^. After a 13.5 F/mol charge, the reaction was stopped, the reaction mixture was evaporated to its half volume under reduced pressure. Then it was moved into a 100-mL beaker and TBAP was precipitated with the addition of diethyl ether (50 mL) and filtered through a cotton pad. The clear, pale brown solution was transferred into a 50-mL round-bottom flask in portions and the solvent was evaporated under reduced pressure. Preparative TLC (hexane–EtOAc 1:2) purification of the brown semi-solid residue gave product **5** (32.2 mg, 41%) as a white solid. The ^1^H NMR spectrum was in accord with literature data [47,48].

## 4. Conclusions

We have investigated the electrochemical mono- and didealkylation of *N,N*-diethylbenzenesulfonamide using a commercially available and a tailor-made electrochemical reactor. The in-depth study shed light on several interesting findings. As regards the choice of solvent, the formation of the methoxy intermediate was preferred in pure methanol, while the presence of the corresponding isopropoxy and *tert*-butoxy congeners was less characteristic in cases where isopropanol or *tert*-butanol was used instead of methanol. The formation of the monodeethylated product became slightly more significant in the presence of 1% water and predominant in solvent mixtures containing 5% water. In some reactions carried out with 5% water, a significant amount of the doubly deethylated derivative was also formed. Further optimization of the electrochemical reaction led to a scaled-up, environmentally benign synthesis of the dealkylated products. To our delight, under appropriate conditions, the corresponding *N*-methoxyalkyl intermediate, a precursor of the *N*-sulfonyliminium cation, could also be isolated.

## Data Availability

Data are contained within the article and Appendix A.

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
