# Peer review of "Detailed Studies on the Methoxylation and Subsequent Dealkylation of N,N-Diethylbenzenesulfonamide Using a Tailor-Made Electrosynthetic Reactor"

_molecules, 2024, doi:10.3390/molecules29235496_

Round 1

Reviewer 1 Report

Comments and Suggestions for Authors

Although alkylation of sulfonamides is a trivial matter, removal of N-alkyl groups (dealkylation) is far less well known. In this paper, an electrochemical method for cleaving one or two alkyl groups from N,N-diethylbenzenesulfonamide is detailed. Different conditions were investigated and in some cases alkoxy derivatives/intermediates were identified. These acetal analogues are potentially useful as protective groups. The authors suggest this approach is environmentally beneficial (green). The study appears to have been carried out with care and there do not appear to be any problems. The writing is mostly clear. The introduction seems a little on the long side but is otherwise fine.

I did not understand Tables 4 and 6 where >99% conversion of 1 occurred but 5-31% 1 was observed by HPLC. Am I missing something?

Minor point: line 77 might be better as "and these can later be captured" rather than "then the latter can be captured".

Line 97: there should be a space after "observed."

Author Response

Dear Reviewer,

on behalf of my co-authors and myself I thank you for your work and your comments. Our responses are listed below, and the manuscript has been amended accordingly.

Comment:

The introduction seems a little on the long side but is otherwise fine.

Response:

The Introduction section has been shortened by the omission of some less essential parts of the text and by displacement of Scheme 1 into the Supporting information.

Comment: I did not understand Tables 4 and 6 where >99% conversion of 1 occurred but 5-31% 1 was observed by HPLC. Am I missing something?

Response: 

After a certain time period (2 h in Table 4 and 5 h in Table 6), a significant decomposition could be observed leading to UV-undetectable components. The area under the UV curve of compound 1 became smaller than before, smaller than 1% of the original area meaning that the conversion was >99%. Nevertheless, due to the intense decomposition of the other products (2, 3 and 5 listed in the table) into UV-undetectable components, the relative amount of 1 compared to that of 2, 3 and 5 started to increase. Thus, the value >99% comes from a comparison with the original spectrum, while the values 5-31% come from a comparison at a certain point of time with the area of other compounds. This phenomenon is mentioned in a sentence above Table 4, the the footers of Table 4 and 6 also refer to this. For the sake of clarity, we have now finetuned the footer of Table 6. 

Comment:

Minor point: line 77 might be better as "and these can later be captured" rather than "then the latter can be captured"

Response:

Thank you for pointing this out. It has been modified to: "these can be captured...".

Comment:

Line 97: there should be a space after "observed."

Response:

Indeed. A space has now been inserted. During the shortening of the introduction section, the preceding sentence has been deleted, so the corresponding part of the manuscript now is as follows: "... (Scheme 2/A) [34]. Shono oxidation ..." 

Reviewer 2 Report

Comments and Suggestions for Authors

The authors have presented a very well written paper discussing their work in an important field.  I recommend publication, and have only a minor comment for the authors.

After the first paragraph of the Introduction, place a ChemDraw or similar structure of the benzenesulfonamide unit separately, before it is shown in the various figures and schemes as a part of the drugs.

Author Response

Dear Reviewer,

on behalf of my co-authors and myself I thank you for your work and your comments. Our response is given below, and the manuscript has been amended accordingly.

Comment:

After the first paragraph of the Introduction, place a ChemDraw or similar structure of the benzenesulfonamide unit separately, before it is shown in the various figures and schemes as a part of the drugs.

Response: 

We have now demonstrated benzenesulfonamide separately, as requested. So that the number of Figures doesn't increase and the figures don't have to be renumbered, the structure of benzenesulfonamide has been inserted into Figure 1 as the very first structural formula of the manuscript.